# Dimethyl Fumarate Protects Retinal Pigment Epithelium from Blue Light-Induced Oxidative Damage via the Nrf2 Pathway

**DOI:** 10.3390/antiox12010045

**Published:** 2022-12-26

**Authors:** Hideyuki Shimizu, Kei Takayama, Kazuhisa Yamada, Ayana Suzumura, Tomohito Sato, Yoshiaki Nishio, Masataka Ito, Hiroaki Ushida, Koji M Nishiguchi, Masaru Takeuchi, Hiroki Kaneko

**Affiliations:** 1Department of Ophthalmology, Nagoya University Graduate School of Medicine, Nagoya 466-8550, Japan; 2Department of Ophthalmology, National Defense Medical College, Tokorozawa 258-8513, Japan; 3Department of Developmental Anatomy and Regenerative Biology, National Defense Medical College, Tokorozawa 258-8513, Japan

**Keywords:** dimethyl fumarate, blue-light, nuclear factor (erythroid-derived)-like 2, retinal pigment epithelium, oxidative stress, NRF2 pathway, age-related macular degeneration

## Abstract

The purpose of this study is to investigate the protective effect of dimethyl fumarate (DMF), the methyl-ester of fumaric acid, against blue-light (BL) exposure in retinal pigment epithelial (RPE) cells. ARPE-19 cells, a human RPE cell line, were cultured with DMF followed by exposure to BL. Reactive oxygen species (ROS) generation, cell viability, and cell death rate were determined. Real-time polymerase chain reaction and Western blotting were performed to determine the change in nuclear factor (erythroid-derived)-like 2 (NRF2) expression. Twenty-seven inflammatory cytokines in the supernatant of culture medium were measured. BL exposure induced ROS generation in ARPE-19 cells, which DMF alleviated in a concentration-dependent manner. BL exposure increased the ARPE-19 cell death rate, which DMF alleviated. BL exposure induced ARPE-19 cell apoptosis, again alleviated by DMF. Under BL exposure, DMF increased the *NRF2* mRNA level and promoted NRF2 expression in the nucleus. BL also strongly increased interleukin (IL)-1β and fibroblast growth factor (FGF) expression. BL strongly induced RPE cell damage with apoptotic change while DMF mainly reduced inflammation in BL-induced RPE damage, resulting in blockade of cell death. DMF has a protective effect in RPE cells against BL exposure via activation of the NRF2 pathway.

## 1. Introduction

Age-related macular degeneration (AMD) is a common cause of visual impairment in the elderly. AMD accounts for about 9% of the world’s blind people. There are approximately 30 million AMD patients, of whom more than 0.5 million are blind. [1]. AMD is produced by oxidative stress-related metabolism and is characterized by the presence of extracellular deposits called drusen that accumulate between the Burch’s membrane and retinal pigment epithelium (RPE). We previously reported that malondialdehyde, an oxidative stress marker, is increased in RPE/choroid of AMD patients compared to controls [2]. AMD may be a multifactorial pathology [3], characterized by photoreceptor cell death [2,4,5,6,7,8]. Known risk factors for AMD include obesity, hypertension, smoking, and light exposure [9,10,11]. Light exposure, especially blue light (BL) exposure, plays an important role in the progression of AMD [12,13,14,15]. BL exposure causes an increase in reactive oxygen species (ROS), resulting in structural damage to the retina and reduced cell viability. Additionally, causes apoptosis of RPE via oxidative stress and mitochondrial damage [16,17,18,19,20]. The nuclear factor (erythroid-derived)-like 2 (Nrf2)/Kelch-like ECH-associated protein 1 (Keap1) signaling pathway is one of the most important antioxidant cellular defense and survival pathways [21]. Nrf2 is a 65 kDa molecule with a basic leucine zipper structure [22]. In the absence of oxidative stress, inactive Nrf2 is bound to Keap1 in the cytoplasm [23]. When cells are exposed to oxidative stress, cysteine residues in the active site of Keap1 are oxidized and the interaction between Keap1 and Nrf2 is inhibited. When Nrf2 accumulates in the cytoplasm, it migrates to the nucleus and binds to antioxidant response elements [24]. Nrf2 also serves as a master regulator of highly coordinated antioxidant responses in RPE cells [25]. Some studies have suggested that the Nrf2/Keap1 signaling pathway protects RPE cells and retinal ganglion cells from BL exposure [26,27,28].

Fumaric acid is found naturally in the plants Fumaria officinalis and mushrooms. Dimethyl fumarate (DMF) is the methyl-ester of fumaric acid and acts as an antioxidant and anti-inflammatory agent. It was used successfully in Germany in the 1950s to treat psoriasis vulgaris [29] and many of the underlying mechanisms of anti-psoriatic action, including a Th1/Th2 balance [30,31,32], inhibition of important leukocyte adhesion molecules [33], the inducing the pro-apoptotic pathway of T cells. [31,34], and, recently, anti-angiogenic effect [35], have been discovered [36]. Moreover, it was found that DMF activates the Nrf2/Keap1 signaling pathway and possesses therapeutic efficiency for some diseases, including alcoholic liver disease, intracerebral hemorrhage, necrobiosis lipoidica, granuloma annulare, and sarcoidosis [37,38,39,40]. In particular, DMF is a first-line-treatment used globally for relapsing-remitting multiple sclerosis [41,42,43]. The mode of action comprises immunomodulatory effects and an activation of Nrf2-mediated antioxidative response pathways leading to additional cytoprotective effects [44,45].

On the basis of the scientific background and current knowledge, we hypothesized that DMF protects RPE cells from BL exposure via the Nrf2 pathway and acts as a new treatment agent for AMD. However, to the best of our knowledge, no studies have focused on the therapeutic potential of DMF as a novel AMD therapy. Therefore, in this study, we investigated the protective effect of DMF against BL exposure in RPE cells.

## 2. Materials and Methods

### 2.1. Cell Culture, Primary Cell Preparation, and BL Exposure

ARPE-19 cells, a human RPE cell line, were purchased from the American Type Culture Collection (Rockville, MD, USA). Cells were grown in colorless Dulbecco’s modified Eagle’s medium premixed with Ham’s F-12 (1:1 ratio, Sigma-Aldrich, St. Louis, MO, USA) and supplemented with 10% fetal bovine serum and the antibiotics streptomycin/penicillin G (Sigma-Aldrich) [2,8]. ARPE-19 cells were exposed to culture solution with DMF (Sigma-Aldrich) for 3 h. Following this exposure, the cells were rinsed with balanced salt solution three times and then immediately cultured in the colorless medium [46] under BL (Zensui LED Lamp BlueTM; Zensui Inc., Settsu, Japan; peak wavelength: 450 µm, with 600 lux, 1000 lux, 1500 lux, 2000 lux).

### 2.2. Lactate Dehydrogenase (LDH) Assay, WST-8 Assay, and ROS Measurement

The cell death rate was evaluated by determining LDH activity using the Cytotoxicity Detection Kit PLUS (Roche Diagnostics, Mannheim, Germany). The supernatant of the culture medium, which contained LDH secreted from dead cells, was collected, followed by the addition of Triton X-100 in the medium to release intracellular LDH from the surviving cells. After measuring the LDH activities in the culture supernatant and medium, the proportion of dead cells among the total cells was calculated. The cell viability on BL exposure was evaluated by the WST-8 assay using the Cell Counting Kit-8 (Dojin East, Tokyo, Japan). RPE cells were seeded in normal growth medium into 96-well cell plates and cultured until the cell density was sufficient. RPE cells were exposed to culture solution with DMF for 3 h, subsequently cultured under BL for 6, 12, 24 h and cell viability was determined. Oxidative stress on BL exposure was evaluated with respect to the amount of ROS, measured using the OxiSelectTM ROS assay kit (Cell Biolabs, Inc, San Diego, CA, USA). In brief, after BL exposure, the assay was terminated by adding cell lysis buffer, and the fluorescence intensity was determined at 493 nm (excitation)/523 nm (emission) using a fluorescence plate reader at each time point.

### 2.3. Cell Morphology and TUNEL Staining

Morphological changes of ARPE-19 cells exposed to BL were visualized using a phase-contrast microscope (FSX-100; Olympus, Tokyo, Japan) and an electron microscope (Type 1400plus; JOEL, Tokyo, Japan). When using the electron microscope, cells were fixed with 2.5% glutaraldehyde and 2% paraformaldehyde in 0.1 M sodium cacodylate (pH 7.4; 1:2 Karnovsky’s fixative) for 15 min, then post-fixed with 2% aqueous osmium tetroxide for 1.5 h. After embedding in Epon 812, ultra-thin sections stained in uranyl acetate were observed under the electron microscope. TUNEL-positive apoptotic cells were detected as previously described [8]. In brief, after 24 h of BL exposure, the cells were fixed with 2% paraformaldehyde for 20 min at room temperature (RT) on the chambered cell culture slides. The cells were stained with the In Situ Cell Death Detection kit (Roche Diagnostics KK, Tokyo, Japan) for 1 h. The stained cells were then observed using a Bio Imaging Navigator fluorescence microscope (BZ-9000; Keyence, Osaka, Japan). The number of TUNEL-positive cells was calculated from images obtained with a 20× lens (537 µm × 710 µm). The mean number of TUNEL-positive cells observed in three independent areas was calculated per well (*n* = number of wells) [8].

### 2.4. RNA Isolation and Quantitative Reverse Transcription-Polymerase Chain Reaction (RT-PCR)

For real time-polymerase chain reaction (PCR) analyses, total RNA was purified using the Qiagen RNeasy Mini Kit (Qiagen, Hilden, Germany) according to the manufacturer’s protocol. RNA concentration and quality were assessed using a Nano Drop ND-1000 spectrophotometer (Nano Drop Technologies, Rockland, DE, USA) [2]. The total RNA was reverse transcribed using the Transcriptor Universal cDNA Master Kit (Roche Diagnostics) starting with 2 μg total RNA from each sample [2]. Real time-PCR was performed using the Thunderbird Probe qPCRMix (Toyobo Life Science, Osaka, Japan) and the Gene Expression Assay containing primers and an FAM dye-labeled TaqMan probe for detecting human NRF2 (HS00965961-g1; Applied Biosystems, MA, USA) and eukaryotic 18S rRNA (Hs 99,999,901 s1; Applied Biosystems) that is available for human 18S rRNA [47]. PCR cycles consisted of a predenaturation step at 95 °C for 2 min followed by 40 cycles of denaturing steps at 95 °C for 15 s and annealing and extending steps at 60 °C for 60 s. The relative expressions of the target genes were determined using the 2^−ΔΔCt^ method.

### 2.5. Western Blotting

Western blotting was performed as previously described [2,8,21]. For total protein collection, the cultured human cells were lysed in RIPA buffer (Sigma-Aldrich) with a protease inhibitor cocktail (Roche Diagnostics, Indianapolis, IN, USA). The lysate was centrifuged at 15,000× *g* for 15 min at 4 °C and the supernatant was collected. Protein concentrations were determined using the TaKaRa BCA Protein Assay Kit (Takara Bio Inc., Shiga, Japan). To measure Nrf2 abundance in the nucleus, ARPE-19 cells were treated with Nuclear Extraction Kit (Abcam, Cambridge, UK), in accordance with the manufacturer’s protocol. Proteins (8 µg) from ARPE-19 cells were run on SDS precast gels (Wako, Osaka, Japan) and subsequently transferred to PVDF membranes. These membranes were washed with PBS-T (0.05 M Tris, 0.138 M NaCl, 0.0027 M KCl, pH 8.0, and 0.05% Tween 20; Sigma-Aldrich) and blocked with 5% nonfat dry milk/PBS-T at RT for 2 h. The membranes were incubated with the rabbit antibody against NRF2 (1:1000; Cell Signaling, Massachusetts, USA) at 4 °C overnight. Total protein loading was assessed by immunoblotting using β-actin (1:1000: Cell Signaling, Danvers, Massachusetts, USA), and nuclear protein loading was assessed by immunoblotting using lamin B (rabbit, 1:1000; Santa Cruz Biotechnology, Santa Cruz, Texas, USA). HRP-linked secondary antibody was used (1:3000, Invitrogen, Massachusetts, USA) at RT for 1 h. The signal was visualized with enhanced chemiluminescence (ECL prime; GE Healthcare, Piscataway, NJ, USA) and captured using a ChemiDoc XRS+ System (Bio-Rad, Hercules, California, USA). NRF2 protein levels were normalized to those of Lamin B1 and β-actin, respectively.

### 2.6. Measurement of Inflammatory Cytokines

After 24 h incubation under BL exposure, 27 inflammatory cytokines in the supernatant of the culture medium were measured using the Bio-Plex multiplex assay (Bio-Plex Human Cytokine 27-plex panel; Bio-Rad, Hercules, CA, USA) and a multiplex bead analysis system (Bio-Plex Suspension Array System; Bio-Rad) according to the manufacturer’s instructions. All standards and samples were assayed in duplicate. Levels of aqueous humor cytokines below detectable levels were considered as 0 for statistical analysis [48].

### 2.7. Outcomes and Statistical Analysis

Data are expressed as mean ± standard deviation (SD) (*n* = number of samples). The data of the cell death rate, ROS generation, real time-PCR, and cytokine levels were analyzed by one-way or two-way analyses of variance (ANOVA) for two components followed by Dunnett’s (to compare the mean of each group with the mean of the target group) or Tukey’s post hoc tests (to compare the mean of each group with the mean of every other group). In all analyses, *p*< 0.05 was considered to indicate statistical significance.

## 3. Results

### 3.1. Cell Death Rate Is Increased by BL Intensity Level and Irradiation Time

First, we investigated how the BL intensity level and irradiation time effected RPE cell death. BL exposure induced ARPE-19 cells to release LDH and BL 1500 lux was found to be the most appropriate for this study (Figure 1A). BL 2000 lux strongly induced cell death whereas BL 600 lux and 1000 lux only weakly caused it. The total LDH of 33.0 ± 8.7% (*n* = 4) at 24 h after BL 1500 lux exposure was significantly higher than that of 13.2 ± 1.0% (*n* = 4) without BL exposure (*p* = 0.0094) (Figure 1B). The irradiation time of 24 h was more appropriate than of 12 h because the BL exposure effect was much greater. The following experiments were performed under conditions of BL 1500 lux exposure.

### 3.2. BL Causes RPE Cell Death, Decreases Cell Viability, and Increases ROS in RPE Cells and DMF Protects RPE Cells from BL

We subsequently examined the protective effect of DMF against BL exposure on ARPE-19 cells in vitro. ARPE-19 cells were exposed to culture solution with DMF for 3 h, cultured under BL for 24 h, and the LDH activity, cell viability, and amount of ROS were determined. BL exposure induced ARPE-19 cells to release LDH, which DMF reduced. LDH was 30.9 ± 1.2% (*n* = 4), 25.0 ± 2.4% (*n* = 4), and 19.1 ± 3.6% (*n* = 4) with 0, 10, and 100 μM DMF, respectively, displaying a dose-dependent effect (0 μM vs. 10 μM and 10 μM vs. 100 μM; *p* = 0.0094 and 0.037, respectively) (Figure 2). By contrast, without BL exposure, the LDH released was 14.2 ± 3.2% (*n* = 4), 13.2 ± 2.0% (*n* = 4), and 13.2 ± 2.8% (*n* = 4) with 0, 10, and 100 μM DMF, respectively, at 24 h, without any dose-dependent effect by DMF treatment (0 μM vs. 10 μM and 10 μM vs. 100 μM; *p* = 1.0 and 0.96, respectively) (Figure 2). The cell viabilities were 49.3 ± 10.5% (*n* = 4), 48.8 ± 13.9% (*n* = 4), and 60.1 ± 6.9% (*n* = 4) with 0, 10, and 100 μM DMF, respectively, at 24 h after BL exposure, this not being concentration dependent (0 μM vs. 10 μM and 10 μM vs. 100 μM; *p* = 1.00 and 0.54, respectively). By contrast, the BL-free/cell viabilities were 100.6 ± 10.7% (*n* = 4), 110.2 ± 6.6% (*n* = 4), and 107.8 ± 3.7% (*n* = 4) with 0, 10, and 100 μM DMF, respectively at 24 h, which were not significantly different from each other (0 μM vs. 10 μM and 10 μM vs. 100 μM; *p* = 0.69 and 1.0, respectively) (Figure 2). The fold-changes in ROS generation were 5.4 ± 1.2 (*n* = 6), 4.0 ± 1.9 (*n* = 6), and 2.0 ± 0.3 (*n* = 6) with 0, 10, and 100 μM DMF, respectively at 24 h after BL exposure, being significantly decreased at the higher DMF concentration (0 μM vs. 10 μM and 10 μM vs. 100 μM; *p* = 0.91 and <0.001, respectively). By contrast, the fold-changes in ROS generation without BL exposure were 1.0 ± 0.3 (*n* = 6), 1.1 ± 0.7 (*n* = 6), and 0.9 ± 0.5 (*n* = 6) with 0, 10, and 100 μM DMF, respectively at 24 h, showing no significant effect of DMF treatment (0 μM vs. 10 μM and 10 μM vs. 100 μM; *p* = 1.0 and 1.0, respectively) (Figure 2).

### 3.3. BL Exposure Induced Morphological Change in ARPE-19 Cell-Associated Apoptosis That DMF Prevented

We examined the morphological change of ARPE-19 cells on BL exposure and whether DMF prevented this using an electron microscope. Under normal conditions, ARPE-19 cells displayed no cell damage using an electron microscope. However, on BL exposure, these cells were induced to undergo apoptosis and there were many vacuoles in the cytoplasm. By contrast, on BL exposure after DMF treatment, these changes were alleviated (Figure 3A). To clarify more precisely the mechanism of ARPE-19 cell death by BL exposure and prevention by DMF treatment, we performed TUNEL staining. The percentage of TUNEL-positive cells was 248 ± 49.4% (*n* = 7) after BL exposure without DMF treatment, 150 ± 46.3% (*n* = 7) with DMF treatment, and 100 ± 12.8% (*n* = 5) without BL exposure nor DMF treatment. The TUNEL-positive cells were significantly increased by BL exposure (*p* = 0.012), and DMF significantly alleviated the percentage of TUNEL-positive cells (*p* = 0.033) (Figure 3B). This staining revealed that numerous ARPE-19 cells were TUNEL-positive at 24 h after BL exposure and this was alleviated by DMF treatment. These findings indicated that BL exposure-induced ARPE-19 cell death mostly involved apoptosis and this was alleviated by DMF treatment.

### 3.4. BL Exposure Increased NRF2 Expression in ARPE-19 Cells and Promoted NRF2 Expression in the Nucleus

To investigate whether DMF effected NRF2 involvement in the response to BL exposure of ARPE-19 cells, we investigated *NRF2* mRNA and protein expression of ARPE-19 cells with or without DMF treatment on BL exposure. Compared with the *NRF2* mRNA level of ARPE-19 cells without DMF treatment with BL exposure (1.28 ± 0.26, *n* = 8), that with DMF (100 μM) treatment with BL exposure was significantly higher (2.14 ± 0.84, *n* = 8; *p* = 0.0019) (Figure 4A). Interestingly, compared with the *NRF2* mRNA level of ARPE-19 cells without DMF treatment without BL exposure (1.02 ± 0.20, *n* = 8), that with DMF (100 μM) treatment without BL exposure was not significantly higher (1.28 ± 0.21, *n* = 8; *p* = 0.70). This implied that, without BL exposure, the *NRF2* mRNA level was not increased by DMF (Figure 4A). As previously we reported [21], compared with NRF2 protein expression in the nucleus of ARPE-19 cells without DMF treatment or BL exposure (1.31 ± 0.23, *n* = 6), that without DMF treatment but with BL exposure was significantly higher (2.02 ± 0.30, *n* = 6; *p* = 0.03). In addition, compared with NRF2 protein expression in the nucleus of ARPE-19 cells without DMF treatment but with BL exposure (2.02 ± 0.30, n = 6), that with both DMF (100 μM) treatment and BL exposure was significantly higher (3.53 ± 0.53, *n* = 6; *p* = 0.02). These results implied that BL exposure promoted NRF2 expression in the nucleus, which was increased further by DMF treatment (Figure 4C). Compared with NRF2 protein expression in the entire ARPE-19 cell without DMF treatment or BL exposure (1.02 ± 0.05, *n* = 6), that without DMF treatment but with BL exposure was not significantly higher (1.04 ± 0.07, *n* = 6; *p* = 0.99). However, compared with NRF2 protein expression in the entire ARPE-19 cell without DMF treatment but with BL exposure (1.04 ± 0.07, *n* = 6), that with both DMF (100 μM) treatment and BL exposure was significantly higher (1.74 ± 0.14, *n* = 6; *p* = 0.02). These results implied that NRF2 protein expression in the entire ARPE-19 cell was increased by DMF. (Figure 4D).

### 3.5. BL Exposure and DMF Altered Cytokines Expression in ARPE-19 Cells

AMD reportedly has a strong relationship with inflammatory cytokines, including vascular endothelial growth factor (VEGF) [2]. Therefore, we measured major inflammatory cytokines secreted from ARPE-19 cells under BL exposure and investigated up- or down-regulation by DMF administration (Table 1). BL exposure increased multiple major pro-inflammatory cytokines, for example, interleukin (IL)-1β, which was reported to be increased in human eyes with AMD [48]. BL also strongly increased fibroblast growth factor (FGF) expression, a key factor that controls fibrotic changes that are reportedly associated with subretinal fibrosis in advanced AMD [49,50]. However, BL did not induce VEGF upregulation, a major proangiogenic factor in AMD. DMF reduced IL-6, a major pro-inflammatory cytokine [51] (Figure 5). Interestingly, DMF did not suppress IL-1β, VEGF, or FGF. These results indicated that DMF partially suppressed inflammation caused by BL exposure in ARPE-19 cells, but did not prevent fibrosis or angiogenesis.

## 4. Discussion

Damage to retinal cells by visible light exposure occurs by type I (free radical) and type II (oxygen-dependent) mechanisms. Free radicals induce cell necrosis and oxygen-dependent mechanisms induce apoptosis [47]. Oxidative stress due to several factors was thought to be involved in the pathogenesis of AMD, and RPE cells were thought to die by apoptosis. BL exposure produces ROS, which damage mitochondrial DNA and cellular structures, resulting in a state of apoptosis in RPE cells [15,16,17]. In the present study, we demonstrated that BL exposure caused ROS generation in RPE cells and induced cell death, which DMF alleviated in a dose-dependent manner. In addition, TUNEL staining indicated that apoptotic RPE cell death on BL exposure was diminished by DMF. BL exposure increased the ARPE-19 cell death rate, which DMF alleviated. However, DMF treatment did not improve ARPE-19 cell viability. These results indicated that DMF can protect ARPE-19 cells from BL exposure and decrease ROS in ARPE-19 cells, but cannot completely restore cell viability. To an extent, these findings simulate the mechanism of the protective role of DMF against BL exposure. In the absence of BL exposure, the NRF2 mRNA level and total NRF2 protein level were not increased, but NRF2 expression in the nucleus was promoted by DMF application. By contrast, under BL exposure conditions, the NRF2 mRNA level and total NRF2 protein expression were increased and NRF2 expression was, furthermore, increased in the nucleus by DMF application. These results suggest that DMF promotes NRF2 translocation from cytoplasm to nucleus and mainly functions protectively in RPE cells against BL exposure via the NRF2 pathway. The anti-oxidative effects of DMF have been reported in various studies using RPE cells in hyperglycemic settings [52,53,54]. Some reports have shown that DMF activates the NRF2 pathway, thus leading to a decrease in intracellular ROS levels. In the same way, BL induces ROS generation and DMF activates NRF2 pathway [53,54]. We performed multiplex analysis to determine any changes in inflammatory cytokines. Among them, we particularly focused on 4 out of 27 inflammatory cytokines on BL exposure with or without DMF. BL strongly increased IL-1β and FGF expression, which was not alleviated by DMF. The advanced form of neovascular AMD displays subretinal fibrosis and, presumably, FGF plays a role in its pathogenesis [49]. However, our results indicated that DMF have a limited role in the prevention of subretinal fibrosis in the advanced form of AMD. Another interesting finding was that BL itself did not induce VEGF upregulation in ARPE-19 cells. Moreover, DMF did not change VEGF expression, the main driver in the pathogenesis of neovascular AMD [2,7,21,55,56]. These results indicate that BL exposure-induced RPE damage and its blockade by DMF do not strongly contribute to the angiogenesis in the pathogenesis of AMD. By contrast, DMF decreased IL-6 expression, a major pro-inflammatory cytokine. DMF might thus contribute to blocking the cytotoxic effect of BL in RPE cells.

Limitations of this study are as follows: (a) We examined only biological changes in vitro. A previous study using mice exposed to direct light exposure in vivo have revealed tissue changes due to oxidative stress [57]. Examining the effects of BL exposure in live mice with and without DMF would provide more accurate information on the adverse effects of BL exposure on the eye and the importance of the NRF2 pathway in protecting the eye during such exposure. (b) There might be a discrepancy in the evaluation of the results between the preventive effect of DMF on the cell death rate and the loss of cell viability. More precise evaluation with additional assays will elucidate the mechanism of the preventive effect of DMF in BL-induced RPE damage. (c) NRF2 protein expression in the entire ARPE-19 cell and the nucleus was evaluated, but that in the cytoplasm was not. To demonstrate NRF2 translocation from cytoplasm to nucleus, we have to show that the NRF2 protein expression in the cytoplasm was decreased by DMF with BL exposure.

## 5. Conclusions

We showed that BL strongly induced RPE cell damage with apoptotic change and increased IL-1β and FGF expression, while DMF mainly reduced inflammation in BL-induced RPE damage, resulting in blockade of increased cell death. DMF had a protective effect in RPE cells exposed to BL by activating the NRF2 pathway.

## Figures and Tables

**Figure 1 antioxidants-12-00045-f001:**
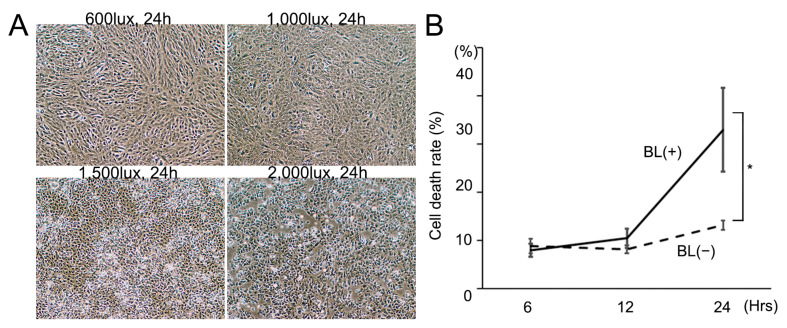
Cell death was enhanced by increasing blue-light (BL) intensity and irradiation time. (**A**) BL exposure increased the cell death rate in an intensity-dependent manner after 24 h exposure. (**B**) The cell death rate was significantly higher with BL 1500 lux exposure than that without BL exposure 24 h later (* *p* = 0.0094; *n* = 4 per group). Statistical analysis was performed using one-way ANOVA, followed by Tukey’s multiple comparisons test where appropriate.

**Figure 2 antioxidants-12-00045-f002:**
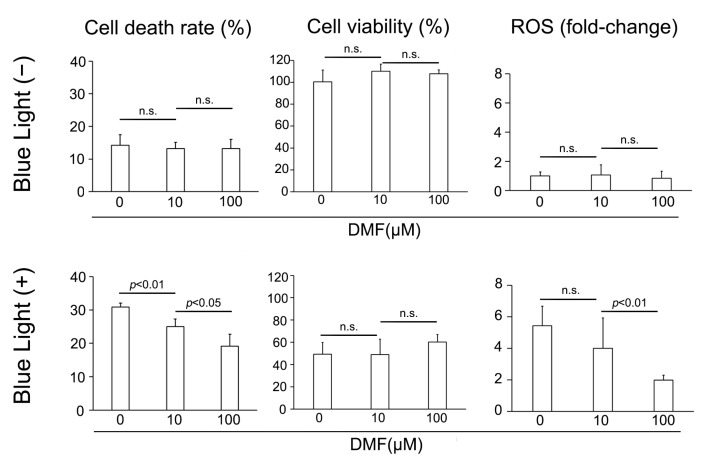
Change in cell death rate, cell viability, and ROS generation after blue-light (BL) exposure and with DMF. BL exposure increased ARPE-19 cell death (LDH release), which DMF decreased it dose-dependent manner (*n* = 4 per group). BL exposure and DMF treatment did not change ARPE-19 cell viability (*n* = 4 per group). BL exposure induced ROS generation in ARPE-19 cells, which DMF alleviated in a dose-dependent manner (*n* = 6 per group). Statistical analysis was performed using one-way ANOVA, followed by Tukey’s multiple comparisons test where appropriate. DMF = dimethyl fumarate, ROS = reactive oxygen species, n.s. = not significant.

**Figure 3 antioxidants-12-00045-f003:**
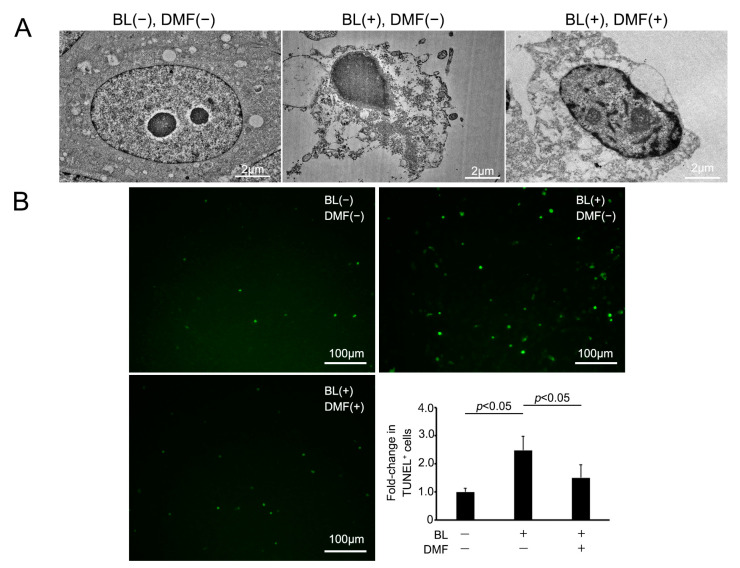
Apoptotic change of ARPE-19 cells after blue-light (BL) exposure and with DMF. (**A**) Following BL exposure, ARPE-19 cells without DMF display numerous vacuoles in the cytoplasm, indicating apoptotic changes. By contrast, those also with DMF only have minor changes. (**B**) TUNEL, an apoptotic cell marker, staining of ARPE-19 cells revealed a significantly greater number of TUNEL-positive cells after 24 h exposure, which was alleviated by DMF (*n* = 7 for BL exposure without DMF treatment and with DMF treatment, *n* = 5 for neither BL exposure nor DMF treatment). Statistical analysis was performed using one-way ANOVA, followed by Tukey’s multiple comparisons test where appropriate. DMF = dimethyl fumarate.

**Figure 4 antioxidants-12-00045-f004:**
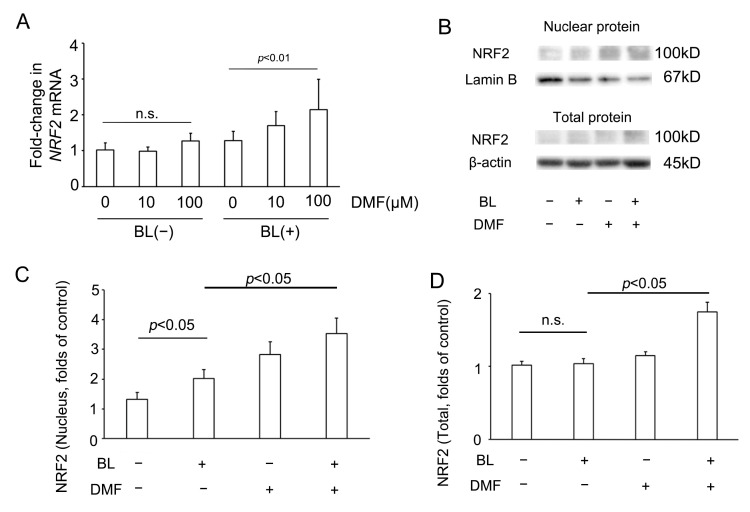
NRF2 expression in ARPE-19 cells on blue-light (BL) exposure. (**A**) In the absence of BL exposure, the NRF2 mRNA level was not increased by DMF. By contrast, on BL exposure, the NRF2 mRNA level was increased by DMF (*n* = 8 per group). (**B**) The nuclear and total protein NRF2 expression, as determined by Western blot. (**C**) The NRF2 protein expression in the nucleus was increased by BL exposure and DMF (*n* = 6 per group). (**D**) In the absence of BL exposure, NRF2 protein expression in the entire ARPE-19 cell was not increased by DMF. By contrast, on BL exposure, the NRF2 protein expression in the entire ARPE-19 cells was increased by DMF (*n* = 6 per group). Statistical analysis was performed using one-way ANOVA, followed by Tukey’s multiple comparisons test where appropriate. NRF2 = nuclear factor (erythroid-derived)-like 2, DMF = dimethyl fumarate, n.s. = not significant.

**Figure 5 antioxidants-12-00045-f005:**
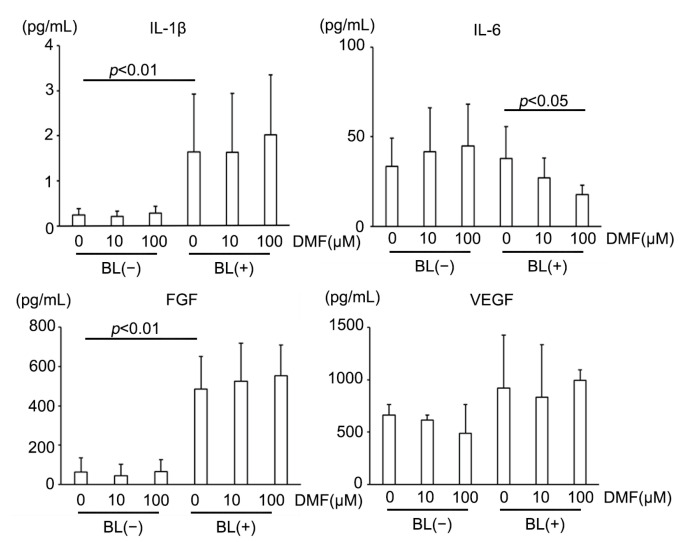
Representative inflammatory cytokine expressions in ARPE-19 cells exposed to blue light (BL). BL strongly increased IL-1β and FGF expression, which DMF did not alleviate (*n* = 10 per group). DMF decreased IL-6 expression (*n* = 10 per group). BL did not increase VEGF expression (*n* = 10 per group). Statistical analysis was performed using one-way ANOVA, followed by Tukey’s multiple comparisons test where appropriate. DMF = dimethyl fumarate, FGF = fibroblast growth factor, VEGF = vascular endothelial growth factor.

**Table 1 antioxidants-12-00045-t001:** Cytokine levels (pg/mL) in the supernatant from ARPE-19 cells treated with DMF under blue light exposure.

		**IL-1β**	**IL-1ra**	**IL-2**	**IL-4**	**IL-5**	**IL-6**	**IL-7**	**IL-8**	**IL-9**
BL (−)	DMF (−)	0.24 ± 0.14	26.92 ± 15.78	5.68 ± 5.93	1.22 ± 1.12	26.87 ± 6.29	33.53 ± 15.60	6.08 ± 3.26	78.35 ± 70.11	4.49 ± 3.18
	DMF 10 μM	0.21 ± 0.12	32.23 ± 18.11	3.72 ± 4.05	1.03 ± 0.98	18.44 ± 6.85	41.80 ± 24.36	5.89 ± 3.88	78.17 ± 50.25	4.86 ± 1.95
	DMF 100 μM	0.28 ± 0.15	18.55 ± 6.64	3.44 ± 3.17	0.95 ± 0.84	16.08 ± 6.80	44.86 ± 23.41	4.31 ± 3.83	66.05 ± 20.30	3.66 ± 2.10
BL (+)	DMF (−)	1.63 ± 1.28	380.72 ± 338.94	17.50 ± 6.04	4.46 ± 1.36	33.39 ± 10.21	37.99 ± 17.74	12.29 ± 4.33	72.13 ± 34.84	10.78 ± 2.05
	DMF 10 μM	1.62 ± 1.31	385.48 ± 367.52	15.93 ± 5.21	4.45 ± 1.40	29.20 ± 5.22	27.21 ± 10.96	6.69 ± 2.98	53.13 ± 22.07	8.67 ± 0.44
	DMF 100 μM	2.00 ± 1.33	472.98 ± 342.23	14.86 ± 5.54	4.91 ± 1.56	25.59 ± 7.80	17.86 ± 5.18	4.57 ± 4.52	56.31 ± 28.77	8.55 ± 1.19
		**IL-10**	**IL-12(p70)**	**IL-13**	**IL-15**	**IL-17**	**Eotaxin**	**FGF**	**G-CSF**	**GM-CSF**
BL (−)	DMF (−)	0.06 ± 0.11	1.17 ± 0.42	0.95 ± 0.08	60.55 ± 71.66	5.05 ± 5.46	3.70 ± 1.88	62.50 ± 74.22	60.96 ± 43.60	4.36 ± 1.08
	DMF 10 μM	0.07 ± 0.10	0.82 ± 0.46	0.44 ± 0.31	43.00 ± 60.80	4.35 ± 4.43	3.65 ± 1.93	45.67 ± 57.60	55.81 ± 32.90	2.46 ± 1.50
	DMF 100 μM	0.12 ± 0.23	1.00 ± 0.49	0.57 ± 0.54	24.53 ± 27.17	3.88 ± 3.94	3.44 ± 1.35	66.04 ± 59.46	56.96 ± 22.13	3.02 ± 1.57
BL (+)	DMF (−)	6.06 ± 9.09	2.82 ± 3.46	2.45 ± 2.12	124.97 ± 95.23	28.21 ± 5.84	5.97 ± 1.75	484.28 ± 167.33	108.66 ± 80.76	4.67 ± 2.36
	DMF 10 μM	3.38 ± 2.47	1.40 ± 1.13	1.77 ± 2.12	111.28 ± 82.12	24.85 ± 6.67	5.57 ± 1.34	524.37 ± 193.48	97.09 ± 72.19	4.11 ± 1.73
	DMF 100 μM	1.72 ± 0.57	0.99 ± 0.13	1.40 ± 1.66	80.73 ± 60.96	25.65 ± 8.43	4.83 ± 1.40	553.46 ± 156.48	86.78 ± 69.68	3.04 ± 1.57
		**IFMγ**	**IP-10**	**MCP-1**	**MIP-1α**	**MIP-1β**	**PDGF-bb**	**RANTES**	**TNFα**	**VEGF**
BL (−)	DMF (−)	205.72 ± 204.71	52.84 ± 31.23	1205.78 ± 676.81	0.36 ± 0.19	2.52 ± 1.93	60.08 ± 32.90	2.36 ± 0.73	45.02 ± 46.59	663.92 ± 97.96
	DMF 10 μM	209.86 ± 167.70	51.41 ± 27.03	1273.79 ± 629.60	0.31 ± 0.10	3.08 ± 1.18	49.44 ± 25.67	2.56 ± 0.91	40.37 ± 34.27	612.72 ± 48.93
	DMF 100 μM	84.21 ± 54.92	52.84 ± 26.46	747.09 ± 110.96	0.29 ± 0.10	1.92 ± 1.02	75.90 ± 44.63	2.90 ± 0.99	36.71 ± 37.35	487.48 ± 276.12
BL (+)	DMF (−)	100.81 ± 67.91	97.09 ± 13.72	661.41 ± 393.39	0.65 ± 0.14	4.89 ± 0.73	70.24 ± 36.67	9.20 ± 6.44	119.53 ± 27.45	919.70 ± 509.56
	DMF 10 μM	72.39 ± 39.10	87.09 ± 13.89	555.43 ± 248.22	0.61 ± 0.16	4.03 ± 0.76	77.70 ± 26.97	6.50 ± 2.72	108.57 ± 18.98	832.50 ± 503.70
	DMF 100 μM	32.40 ± 11.02	87.43 ± 16.21	277.02 ± 102.51	0.64 ± 0.12	4.11 ± 0.75	70.92 ± 21.07	6.85 ± 3.63	102.93 ± 30.99	995.55 ± 99.34

Mean ± SD (pg/mL), BL = blue light, DMF = dimethyl fumarate.

## Data Availability

Data used to support the findings of this study are available from the corresponding author upon request. (email: h-kaneko@med.nagoya-u.ac.jp).

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
