# Peer review of "Dimethyl Fumarate Protects Retinal Pigment Epithelium from Blue Light-Induced Oxidative Damage via the Nrf2 Pathway"

_antioxidants, 2022, doi:10.3390/antiox12010045_

Round 1

Reviewer 1 Report

The authors presented a nice study about the protective effects of DMF on BL-induced ARPE cell damage. The premise of the study was scientifically sound, and the experiments were well conducted. The anti-oxidative effects of DMF have been reported in various studies using RPE cells in hyperglycemic settings. Further, the study lacks in vivo models; thus, the knowledge gained from the current study is minimal.

Author Response

We thank the reviewer for the comment that the premise of the study was scientifically sound, and the experiments were well conducted.

The study lacks you pointed out was the correct and has been written it in the discussion section on page 11, line456;

“Limitations of this study are as follows: (a) We examined only an in vitro biological change. A previous study in mice subjected to direct light exposure in vivo revealed tissue change caused by oxidative stress [58]. Studying the effect of BL exposure on living mice with or without DMF will provide more precise information regarding the adverse effects of BL exposure on the eyes and the importance of the NRF2 pathway in protecting eyes during such exposure.”

And we added the contents about the anti-oxidative effects of DMF in hyperglycemic settings on page 10, line 434;.

“The anti-oxidative effects of DMF have been reported in various studies using RPE cells in hyperglycemic settings [53-55]. Some reports have shown that DMF activates the NRF2 pathway, thus leading to a decrease in intracellular ROS levels. In the same way, BL in-duces ROS generation and DMF activates NRF2 pathway [54-55].”

Reviewer 2 Report

The authors studied protective potential of dimethyl fumarate on ARPE19 cell culture followed by exposure to blue light. They showed that DMF decreased  cell death rate, tunel+ cell, ROS and IL-6 level when cells were treated by BL. The manuscript is written concisely and neatly, without unnecessary information.

The work can be accepted for publication after clarification of the following comments.

1) Specify the protein isolation method for different cell fractions

2) Present data as M±SD (not SE!). Current standards of high impact Q1 journals prefer SD or IQR, not SE.

3) Specify in the legend for each figure n of experiments and statistical method

4) Indicate the molecular weight of proteins in the figure 4. I'm a little confused by the nature of the western blot strip. Could you send the full blot?

5) In abstract: “Under BL exposure, DMF increased the NRF2 mRNA level and  promoted NRF2 translocation from the cytoplasm to the nucleus”. I don't think you can claim that a protein has translocated to the nucleus from blot data alone. Especially since you did not examine the cytosolic fraction. It may be necessary to soften the stated provisions

Author Response

We thank the reviewer for recognizing that our submitted article interesting. We recognize that there are a number of significant concerns that need to be improved in our submitted article. We revised our manuscript following the reviewers’ suggestion.

#2-1: Specify the protein isolation method for different cell fractions.

We specified the protein isolation method in methods section on page 3, line 135;

“For total protein collection, the cultured human cells were lysed in RIPA buffer (Sig-ma-Aldrich) with a protease inhibitor cocktail (Roche Diagnostics, Indianapolis, IN, USA). The lysate was centrifuged at 15,000 ×g for 15 min at 4°C and the supernatant was col-lected. Protein concentrations were determined using the TaKaRa BCA Protein Assay Kit (Takara Bio Inc., Shiga, Japan). To measure Nrf2 abundance in the nucleus, ARPE-19 cells were treated with Nuclear Extraction Kit (Abcam, Cambridge, UK), in accordance with the manufacturer’s protocol.”

#2-2: Present data as M±SD (not SE!). Current standards of high impact Q1 journals prefer SD or IQR, not SE.

Thank you for your advice. We have changed all data as M±SD not SE.

#2-3: Specify in the legend for each figure n of experiments and statistical method.

We specified in the legend for each figure n of experiments and statistical method.

#2-4: Indicate the molecular weight of proteins in the figure 4. I'm a little confused by the nature of the western blot strip. Could you send the full blot?

We indicated the molecular weight of protein in the Figure 4. We have already the nature of the western blot strip in Non-published Material file. We attach it to our reply.

#2-5: In abstract: “Under BL exposure, DMF increased the NRF2 mRNA level and promoted NRF2 translocation from the cytoplasm to the nucleus”. I don't think you can claim that a protein has translocated to the nucleus from blot data alone. Especially since you did not examine the cytosolic fraction. It may be necessary to soften the stated provisions.

That is as you pointed out. We soften the stated provisions and changed sentences in Abstract on page 1, line 23;

“Under BL exposure, DMF increased the NRF2 mRNA level and promoted NRF2 expression in the nucleus.”

And changed sentences in the Results section on page 7, line 357;

“These results implied that BL exposure promoted NRF2 expression in the nucleus, which was increased further by DMF treatment (Figure 4C).”

Thank you.

Round 2

Reviewer 2 Report

The paper can be accepted.